# Integrated Approaches to Identify miRNA Biomarkers Associated with Cognitive Dysfunction in Multiple Sclerosis Using Text Mining, Gene Expression, Pathways, and GWAS

**DOI:** 10.3390/diagnostics12081914

**Published:** 2022-08-08

**Authors:** Archana Prabahar, Kalpana Raja

**Affiliations:** 1R&D Division, Eriks-Precision Components India Pvt Ltd., Mohali 160055, Punjab, India; 2Regenerative Biology, Morgridge Institute for Research, Madison, WI 53715, USA

**Keywords:** multiple sclerosis, cognition, genes, miRNA, GWAS signals, pathways, text mining

## Abstract

Multiple sclerosis (MS), a chronic autoimmune disorder, affects the central nervous system of many young adults. More than half of MS patients develop cognition problems. Although several genomic and transcriptomic studies are currently reported in MS cognitive impairment, a comprehensive repository dealing with all the experimental data is still underdeveloped. In this study, we combined text mining, gene regulation, pathway analysis, and genome-wide association studies (GWAS) to identify miRNA biomarkers to explore the cognitive dysfunction in MS, and to understand the genomic etiology of the disease. We first identified the dysregulated miRNAs associated with MS and cognitive dysfunction using PubTator (text mining), HMDD (experimental associations), miR2Disease, and PhenomiR database (differentially expressed miRNAs). Our results suggest that miRNAs such as hsa-mir-148b-3p, hsa-mir-7b-5p, and hsa-mir-7a-5p are commonly associated with MS and cognitive dysfunction. Next, we retrieved GWAS signals from GWAS Catalog, and analyzed the enrichment analysis of association signals in genes/miRNAs and their association networks. Then, we identified susceptible genetic loci, rs17119 (chromosome 6; *p* = 1 × 10^−10^), rs1843938 (chromosome 7; *p* = 1 × 10^−10^), and rs11637611 (chromosome 15; *p* = 1.00 × 10^−15^), associated with significant genetic risk. Lastly, we conducted a pathway analysis for the susceptible genetic variants and identified novel risk pathways. The ECM receptor signaling pathway (*p* = 3.98 × 10^−8^) and PI3K/Akt signaling pathway (*p* = 5.98 × 10^−5^) were found to be associated with differentially expressed miRNA biomarkers.

## 1. Introduction

Multiple sclerosis (MS) is a long-lasting autoimmune disease of the central nervous system (brain and spinal cord) that leads to demyelination and neurodegeneration [1]. Nearly 2.5 million patients are suffering with MS worldwide, and individuals in the age group of 20 to 40 are the most affected [2]. MS is termed as a medical mystery because the cause of the disease remains elusive. A common assumption is that MS is caused by an abnormal immune response to ecological factors in individuals with a genetic predisposition [3]. A neurological manifestation of MS comprises motor and visual deficits, limb weakness, sensory problems, gait ataxia, uncontrolled bladder, and cognitive decline [4,5]. In addition to the known motor and sensory problems, the effect of cognitive dysfunction is very common in more than half of all the individuals with MS [6,7].

Apart from MS, cognitive dysfunction is a common condition in patients with neurodegenerative diseases. It causes focal impairment to brain structures mediating mild to moderate cognition, and it affects personality, behavior, and decision-making abilities [8,9]. Cognitive dysfunction in MS patients specifically includes declined information processing, memory, concentration, reasoning, and verbal fluency [10,11]. The pathological characteristics of MS include multifocal lesions, inflammation, loss of oligodendrocyte, various ranges of axonal injury, gliosis, and temporary dysfunction to permanent loss [12].

In our study, we aimed to identify biomarkers in MS cognitive impairment analysis using miRNA pathway associations involved in MS cognition. Biomarkers play an important role in the primary assessment and evaluation of a disease, which helps in early detection and treatment. They act as the indicators of changes in normal biological processes and pathological processes, helping to extend our knowledge about the underlying disease conditions [13]. Biomarkers identify neurological disease at an early stage and help to determine the disease condition. If the disease worsens or improves in the patient, the concentration of the biomarker might increase or decrease accordingly; hence, biomarkers act as a noninvasive method of detection for the patients [14].

Gaetani et al. investigated cerebrospinal fluid biomarkers and determined different neuropsychological patterns of cognitive deficits in MS [15]. The cerebral hypoperfusion in MS is associated with chronic hypoxia, focal lesion formation, diffuse axonal degeneration, cognitive dysfunction, and fatigue. MS patients with pathological vascular abnormalities may have reduced quality of life and worsened cerebral perfusion, which may contribute to greater disease progression. Patients with cerebrovascular diseases have poor cognitive performance in MS. These findings suggest that restoring the cerebral blood flow may act as a new therapeutic target in MS [16,17].

The identification of disease-specific biomarkers provides an opportunity to the patients to start the treatment early enough to delay the disease progression [11]. Examples of biomarkers in the progression of MS are (i) MxA expression defining the pathological effect of interferon-β (IFN-β) in vivo [18], (ii) TNF-related apoptosis-inducing ligand (TRAIL), a potential biomarker of IFN-β therapeutic efficacy [19], (iii) complement regulator factor H, a serum biomarker predicted to be effective in disease progression [20], and (iv) the genes CHI3L1 and NF-L, recently identified as biomarkers for mild cognitive impairment in early stages of MS at genomic levels [21]. MicroRNAs (miRNAs) play an important role in various cellular functions and disease-causing mechanisms [22,23]. miRNAs are known to play a critical role in neuron development and maintenance of the central nervous system, including neural differentiation, synaptic plasticity, and cognitive functions [24,25].

An MS cognition study performed by Liguori et al. demonstrated the expression of miRNA in association with cognitive impairment in pediatric MS and reported the associated genes, BST1, NTNG2, SPTB, and STAB1 [26]. Due to the complexity of the disease, various clinical phenotypes are observed in patients at different stages of MS, which led the researchers to analyze various biological and pathological pathways responding to the therapy at both genomic and transcriptomic levels [27,28]. Despite the occurrence of cognitive symptoms in MS, the role of miRNAs and their target mRNAs associated with cognitive decline and neuropsychiatric problems in MS disease is less reported. A study by Scapoli et al. highlighted the influence of vascular components in MS. The study combined the transcriptome analysis of the internal jugular vein (IJV) walls from MS patients and whole-exome sequencing (WES) [29]. Studies on the expression patterns of the IJV wall in MS through combined transcriptome–protein analysis highlighted the proteins of interest for MS pathophysiology [30].

GWAS and other mapping studies have identified 250 loci for autoimmune diseases, some of which are common to two or more diseases. Most of the variants are in the noncoding regions, especially in miRNA target sites [31]. miRNAs act by imperfect base paring with target messenger RNA (mRNA), leading to the negative regulation of mRNA. Single-nucleotide polymorphisms (SNPs) in mRNA sites complementary to miRNA are referred to as miRSNPs [32]. miRSNPs have functional consequences for autoimmune diseases including MS [31]. Statistical approaches and meta-analysis of neurological diseases A recent study showed that miRSNPs have an expression quantitative trait locus (eQTL) effect on genes reported in GWAS [33], and the number of miRSNPs associated with human diseases keep increasing [34,35]; A recent study integrated GWAS with pathways to report novel risk pathways and genes [36]. We believe that the existing analysis on MS cognitive dysfunction can reveal the pathways and genes with common risks that are associated with the cognitive dysfunction in MS. This study can clarify the disease mechanism and its genetic etiology.

This paper extracted the biomarkers for various cognitive function changes in MS patients reported by high-throughput experiments and other experimental techniques from PubMed abstracts using PubTator [37], a text-mining tool. Cognitive problems occurring due to alterations in normal functions of memory, attention and concentration, information processing, executive function, and verbal fluency that are associated with cognitive dysfunction, cognitive impairment, cognitive decline, etc. are reported here. MS cognition and potential disease associations were identified for a wide range of clinical use. While many MS cognition-associated biomarkers have been reported in recent years, this is the first attempt to present an integrated approach to study all the genes, miRNAs, and associated pathways.

## 2. Materials and Methods

### 2.1. Data Collection

We extracted the research articles from PubMed (https://www.ncbi.nlm.nih.gov/pubmed, accessed on 30 June 2019) using the search terms multiple sclerosis and cognition: “multiple sclerosis” [MeSH terms] OR (“multiple” [all fields] AND “sclerosis” [all fields]) OR “multiple sclerosis” [all fields]) AND (“cognitive” [all fields] OR “cognition” [all fields]) AND “humans” [MeSH terms]. We retrieved 4102 publications (June 2019) and exported the PubMed IDs of retrieved articles to a text file for further processing. The retrieved articles were used as input for text-mining analysis.

### 2.2. Gene Concept Finding Using PubTator

PubTator [37] supports five kinds of biological concepts: gene, disease, chemical, species, and mutation. We utilized the gene concept finding, a module within PubTator, for recognizing miRNA and gene mentions in PubMed articles. (https://www.ncbi.nlm.nih.gov/research/bionlp/APIs/usage/, accessed on 30 June 2019) [38]. The gene concept finding module contains information on genes (and gene IDs) reported in each PubMed article. Text-mining results from PubTator were retrieved using libwww-*perl* (LWP), a Perl script. The retrieved information was manually verified, and a nonredundant set of data was mapped to Entrez gene name. For each gene, the official gene symbol approved by the HUGO gene nomenclature committee (HGNC) [39] was retrieved from the NCBI database [40]. 

### 2.3. Predicting Gene–miRNA Targets

We used DIANA-TarBase v8.0, a database which provides high-quality manually curated experimentally validated miRNA–gene interactions [41], to determine the experimentally validated miRNA–gene interactions. The database includes information on the methodology, experimental conditions such as cell/tissue type, and study design. The results are enhanced with detailed metadata. The genes associated with MS cognition in human were used to determine the corresponding miRNA targets and the associated mRNAs. A single miRNA can bind to hundreds of target mRNAs, or a single mRNA can be targeted by multiple miRNAs. These miRNA–mRNA associations play an important role in the regulation of genome [42,43,44].

### 2.4. Identification and Analysis of Genetic Risk Variants

GWAS Catalog [45] was used to identify the risk variants associated with MS and cognitive dysfunction. For MS, we used the keyword “multiple sclerosis”. For cognitive dysfunction, we used two keywords, “cognitive impairment” (CI) and “cognitive decline” (CD). A list of 458 associated genetic variants were retrieved for MS, 150 associated genetic variants were retrieved for CI, and 33 associated genetic variants were retrieved for CD. We selected 77 genetic variants for MS and 25 genetic variants for CI with a *p*-value less than 1 × 10^−5^. For MS, the genetic variants were associated with MS, MS pleiotropy, MS neuromyelitis optica, and MS cholesterol measurements. For CI, the genetic variants were associated with cognitive decline rate in late mild cognitive impairment. For CD, no genetic variants with a *p*-value less than 1 × 10^−5^ were retrieved.

### 2.5. Function Analysis

HaploReg tool (v4.1) [46] was used to annotate the genetic variants, 77 for MS and 25 for CI, and to identify both coding and noncoding mutations. PhenoScanner (v2) [47] database, a collection of human genotype–phenotype associations, was used to perform eQTL analysis on the selected genetic variants. In addition, we used miRDSNP (mirdsnp.ccr.buffalo.edu), a database of disease-associated SNPs and microRNA target sites on 3’UTRs of human genes and MSDD (MiRNA SNP Disease Database, http://bio-bigdata.hrbmu.edu.cn/msdd/, accessed on 15 July 2019) [48] to determine miRNA.

### 2.6. Pathway Analysis

DIANA-mirPath (mirPath v.3) [49] is a webserver for miRNA pathway analysis that utilizes KEGG (Kyoto Encyclopedia of Genes and Genomes) pathways for analysis. This webserver utilizes experimentally validated miRNA interactions derived from DIANA-Tarbase database [41]. In this study, we used a combined list of MS cognition genes by using overrepresentation statistical analysis. This enrichment analysis identified the pathways significantly enriched with genes targeted by at least one of the selected miRNAs. The network was visualized using Cytoscape Software 3.7.2 [50]. This study used the cytoHubba plugin in Cytoscape to determine the potential association of pathways to genes associated with MS cognitive dysfunction [51]. The key genes responsible for MS cognitive dysfunction were identified by the cytoHubba plugin and miRNA–gene network.

### 2.7. Functional Annotation

We performed functional annotation in two phases, functional annotation of genes from text mining and other databases, and identification of risk pathways from the list of risk variants from GWAS. We used DAVID, a Database for Annotation, Visualization, and Integrated Discovery (https://david.ncifcrf.gov/, accessed on 1 August 2019), for identifying the functional annotation of MS cognition genes. The input to DAVID was a list of official gene symbols with a *p*-value cutoff of 0.05. We identified key biological processes (BP), molecular functions (MF), and cellular components (CC) using DAVID [52,53]. For this analysis, *p* < 0.05 was considered to be statistically significant, and the calculated false discovery rate (FDR) score was taken for the study.

## 3. Results

### 3.1. Collection of MS Cognition Genomic and Transcriptomic Data from Literature

PubTator identified 16 genes and 20 miRNA mentions in 4102 PubMed articles retrieved for MS and cognition. A total of 318 gene–miRNA associations were retrieved using the gene concept identification module of PubTator. All these associations were manually curated and verified to get an accurate list of genes/miRNAs. A nonredundant list of genes/miRNA and PubMed IDs were collected using an in-house Perl program. The genes associated with MS cognition were further mapped to their official gene symbols approved by HGNC, and a set of 16 genes were found to be involved in the cognition. A set of 20 nonredundant miRNAs associated with cognitive dysfunction in MS were identified and mapped to miRNA IDs from miRBase [54]. The identified list of genes and miRNAs with PubMed reference is shown in Appendix A (gene-pubmed).

### 3.2. Identification of miRNA–mRNA Targets

Using DIANA-TarBase v8.0, experimentally validated miRNA–gene interactions were identified for each of the gene entity from our list of 16 genes. A total of 248 miRNA–mRNA associations were found along with the experimental evidence and related PubMed resource. This miRNA–mRNA pairs contained 83 nonredundant miRNAs, and these were visualized using Cytoscape software. The list of miRNA–mRNA interactions is shown in Appendix A (gene-miRNA).

### 3.3. Functional Analysis of Genetic Variants

Functional analysis of genetic variants using Haploreg tool identified 63 eQTLS from MS and one eQTL associated with CI. A total of 57 coding and 19 noncoding mutations were found in MS, while 14 coding and 11 noncoding mutations were found in CI. Among the noncoding mutations, 16 were novel transcripts and three were long noncoding RNAs in MS, while eight were novel transcripts and three were long noncoding RNAs in CI. Using PhenoScanner, we identified that 77 genetic variants were eQTLs in MS, and intronic and intergenic locations were identified. The detailed descriptions are shown in Appendix A for MS and Appendix A for CI. Allelic variations and more detailed information are also provided.

### 3.4. KEGG Pathway Analysis

miRNAs involved in MS cognition dysfunctions were subjected to pathway analysis using DIANA-mirPath (mirPath v.3). KEGG pathway analysis utilizes overrepresentation statistical analysis and determines the experimentally validated results with a *p*-value. The network was visualized using Cytoscape 3.2, and the cytoHubba plugin was used to identify the hub genes associated with the reported pathways [55]. The miRNA–pathway associations are shown in Appendix A. The complete list of 185 miRNA pathway interactions is given in Appendix A (miR-KEGG).

Our results show the most significant pathways identified using the highest degree of distribution of the nodes. hsa-mir-155-5p has a higher number of pathway interactions; hence, this miRNA entity may serve as the main target for MS cognitive dysfunction therapy and treatment. hsa-mir-182-5p, hsa-mir-320a, hsa-mir-148b-3p, and has-mir-301a-5p also serve as the biomarkers in MS cognitive impairment analysis. Further analysis of these miRNA pathway interactions determined the most significant pathways, as shown in Figure 1. The ECM receptor pathway is highly involved in the miRNA pathway network.

### 3.5. Identification of KEGG Pathways Based on Genetic Variants from GWAS

Using these 76 mapped genes of MS, we identified 100 pathways associated with MS. Using 26 mapped genes of these genetic variants in CI, we identified 88 pathways associated with CI. These pathways were determined using KEGG pathway mapper https://www.genome.jp/kegg/tool/map_pathway1.html/, accessed on 15 August 2019). From these results, we determined that the genes mapped by the genetic variants are also involved in the ECM signaling pathway, PIK3/Akt signaling pathway, estrogen signaling pathway, and TGF-beta signaling pathway. In addition, we performed the functional annotation of the mapped genes of MS and CI.

### 3.6. DAVID Functional Annotation

All the target genes in the list were given as input to the DAVID tool to identify the related KEGG pathways. Gene Ontology (GO) analysis results revealed that the target genes were expressively enriched in biological processes (BP), including regulation of neurological system process, defense response, regulation of neuron apoptosis, regulation of synaptic transmission, regulation of signaling, neuron development, etc., as shown in Table 1. Molecular functions (MF) in GO included cytokine activity, growth factor activity, neurotrophin receptor binding, interferon-alpha/beta receptor binding, tumor necrosis factor receptor superfamily binding, and structural constituent of cytoskeleton, as shown in Table 2. Cellular components (CC) in GO involved the extracellular region, extracellular region part, and extracellular space functions, as shown in Table 3. Data with the complete list of reported GO functional annotation are shown in Appendix A. The genes involved in each of GO term functional enrichment analysis reflect the enriched genes involved in various biological pathways, as shown in Figure 2.

The functional annotation of 63 mapped genes from MS genetic variants and 26 mapped genes from CI genetic variants was determined using the DAVID tool. The results of KEGG enrichment analysis and GO pathway analysis are shown in Appendix A for MS and Appendix A for CI.

### 3.7. Identification of Key Risk Genes in miRNA–Gene Network

The miRNA–mRNA interactions involving 16 genes and 83 miRNAs were studied to determine the most significant genes that may act as risk genes in the complete MS cognition network. Among these interactions, brain-derived neurotrophic factor (BDNF) was the most significant gene, followed by interferon beta (IFNB), neurofilament L (NEFL), neurotropin (NTF3), and interleukin 10 (IL10). These genes were found to be the top five nodes ranked by cytoHubba, as shown in Figure 3. Other important genes included nerve growth factor (NGF) and chitinase-3-like protein 1 (CH3L1).

## 4. Discussion

We presented an integrated bioinformatics approach to analyze and report the most significant genes that could act as biomarkers in the treatment of MS cognitive impairment. Our approach used text mining, GWAS signals, and pathway analysis to identify the biomarkers associated with MS. Although several genomic and transcriptomic studies are currently reported in MS cognitive impairment, a comprehensive repository dealing with all the experimental data is still underdeveloped. Automated approaches like our approach will be useful to identify the biomarkers associated with MS cognitive impairment in an efficient way.

Cognitive impairment is known to be involved in the mTOR signaling pathway [56,57], PI3K/Akt/mTOR signaling pathway [58], cholesterol biosynthesis pathway [59,60], cholinergic pathway [61,62,63], visual pathway [64], etc. Management of cognitive dysfunction and decline in MS patients is more important to improve the lifestyle of people with MS. Several clinical trials reported the use of simvastatin in MS (MS-STAT trial), affecting brain atrophy, clinical, and cognitive outcome measures. These studies implicate the involvement of metabolites in the cholesterol synthesis pathway.

The development of computational models might elucidate the causal architecture underlying the treatment effects in clinical trials of progressive MS. Physical activity is associated with the motor function and mobility of healthy people, and MS patients have poor physical activity. MS and aging predict the motor function in people, but their cognitive function is not modified. Hence, there is huge need to evaluate and treat cognitive dysfunction in these patients who suffer from MS.

Studies suggest that visual pathway measures are known to bring improved outcomes in MS cognition during neuroprotection trials [65,66]. MS cognition is associated with the cholinergic pathway, but studies suggest that MS does not involve the reduction of cholinergic neurons as in other diseases like Alzheimer’s [67]. Acetyl choline esterase (AChE) inhibitors are used to treat cognitively impaired MS patients, but their therapeutic effects are poor and unsatisfactory [67,68].

CI genetic variants determined from GWAS signals reveal the presence of eQTLs that could regulate the expression of genes with *p* < 0.01, e.g., mapping gene PARP6 with the variant rs11637611 and risk allele C. This study was earlier reported by Hu et al. (rs11637611 with SNPs at chr15q23 and locus *p* = 1.07 × 10^−15^ [69]. Risk genes associated with CI are involved in risk pathways such as the TGF-beta signaling pathway, ECM receptor interaction, PI3k/Akt signaling pathway, mTOR signaling pathway, and estrogen signaling pathway. These results are in correlation with gene/miRNA functional analysis, thus confirming that these risk genes are highly associated with MS cognitive dysfunction.

Risk variant rs703842 located on chromosome 12 was found to be in association with CYP27B1 with a *p*-value < 1.00 × 10^−5^. However, this association was earlier reported by Jiang et al. in the association between genetic polymorphism rs703842 in CYP27B1 and MS [70]. Although MS genes were associated with cognition genes, the genes reported by GWAS variants in MS and CI were not correlated. When functional annotation was performed on these genes, we found that their pathways were correlated (see Appendix A).

In addition to the identification of risk genes from GWAS, we identified miRSNPS from the MSDD database [47] and found that hsa-miR-223 and hsa-miR-23a were associated with genetic variants such as rs1044165 and rs3745453, respectively [70]. These associations were reported in the literature [70,71]. However, our GWAS analysis did not validate these results. Variants such as rs17119 and rs1843938, located on chromosomes 6 and 7, were mapped with novel transcripts with a *p*-value of 1.00 × 10^−10^. These variants were highly correlated. Their associations were earlier validated by GWAS studies performed by Disanto et al. [72].

## 5. Conclusions

Our study showed that BDNF, IFNB, IL10, NEFL, and CHI3L1 could be the potential core genes that play an important role in cognitive dysfunction and impairment in MS. GWAS signals identified the risk variants and alleles associated with MS and cognition, as well as their association with the mapped genes and risk pathways. To further confirm these studies, we utilized miRNAs and their target genes to determine the risk pathways associated with miRNAs. The results from both miRNA functional annotation and GWAS risk gene functional annotation were correlated to determine the key functional genes and risk pathways. In the future, we will evaluate our findings using gene expression datasets in MS patient samples who are cognitively impaired or have cognitive decline.

## Figures and Tables

**Figure 1 diagnostics-12-01914-f001:**
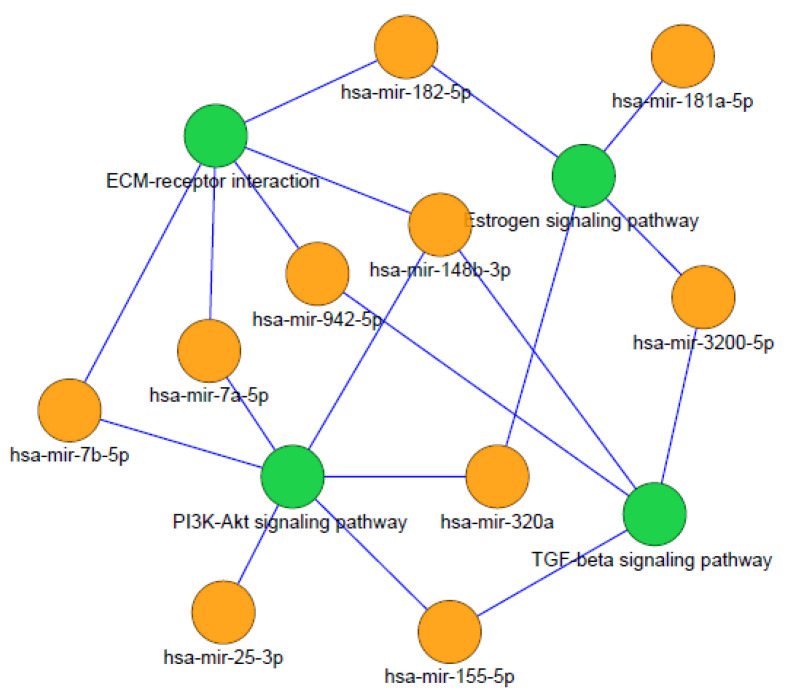
miRNA pathway interactions of most significant pathways.

**Figure 2 diagnostics-12-01914-f002:**
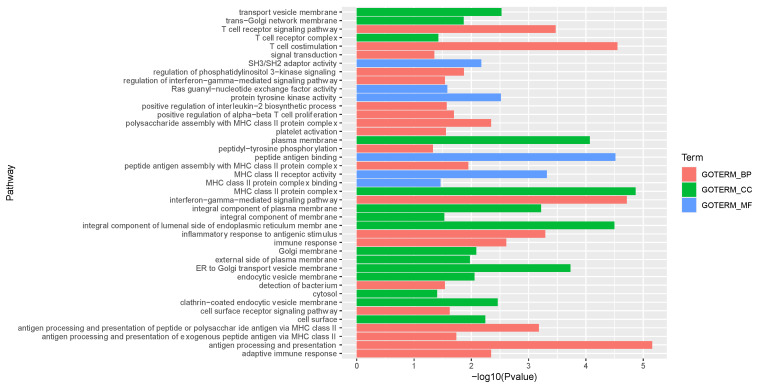
The *X* and *Y*-axes represent pathways (GO Terms) and the −log_10_ *p*-value of the corresponding GO term, respectively. GO terms are represented as follows: BP (biological process), CC (cellular component), and MF (molecular function).

**Figure 3 diagnostics-12-01914-f003:**
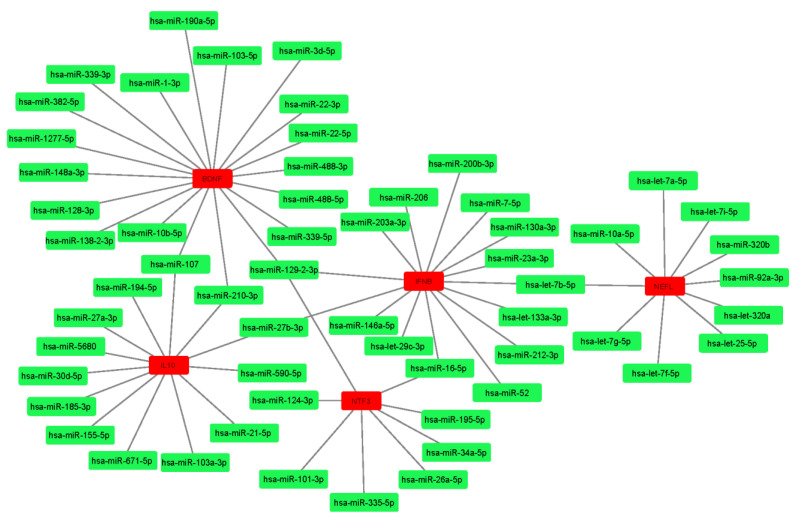
Top five ranked gene nodes and their interactions from gene–miRNA network.

**Table 1 diagnostics-12-01914-t001:** GO analysis biological process (BP) of genes associated with cognitive dysfunction in MS.

GO ID	Description	Count	*p*-Value	Genes
GO:0031644	Regulation of neurological system process	6	4.75 × 10^−7^	BDNF, TNF, NTF4, NTF3, IL10, NGF
GO:0006952	Defense response	8	1.81 × 10^−6^	IL17A, IFNA1, TNF, STAB1, IFNB1, IL17F, IL10, NGF
GO:0043523	Regulation of neuron apoptosis	5	2.36 × 10^−6^	BDNF, TNF, NTF3, NEFL, NGF
GO:0050804	Regulation of synaptic transmission	5	1.22 × 10^−5^	BDNF, TNF, NTF4, NTF3, NGF
GO:0009611	Response to wounding	7	1.30 × 10^−5^	IL17A, TNF, STAB1, IL17F, NEFL, IL10, NGF
GO:0044057	Regulation of system process	6	1.50 × 10^−5^	BDNF, TNF, NTF4, NTF3, IL10, NGF
GO:0051969	Regulation of transmission of nerve impulse	5	1.66 × 10^−5^	BDNF, TNF, NTF4, NTF3, NGF
GO:0006954	Inflammatory response	6	1.91 × 10^−5^	IL17A, TNF, STAB1, IL17F, IL10, NGF
GO:0009617	Response to bacterium	5	4.85 × 10^−5^	TNF, STAB1, IFNB1, IL10, NGF
GO:0051384	Response to glucocorticoid stimulus	4	7.98 × 10^−5^	TNF, NEFL, IL10, NGF
GO:0031960	Response to corticosteroid stimulus	4	1.03 × 10^−4^	TNF, NEFL, IL10, NGF
GO:0042981	Regulation of apoptosis	7	1.36 × 10^−4^	BDNF, TNF, NTF3, IFNB1, NEFL, IL10, NGF
GO:0043067	Regulation of programmed cell death	7	1.44 × 10^−4^	BDNF, TNF, NTF3, IFNB1, NEFL, IL10, NGF
GO:0031175	Neuron projection development	5	1.45 × 10^−4^	BDNF, NTF3, NTNG2, NEFL, NGF
GO:0010941	Regulation of cell death	7	1.47 × 10^−4^	BDNF, TNF, NTF3, IFNB1, NEFL, IL10, NGF
GO:0051094	Positive regulation of developmental process	5	1.99 × 10^−4^	BDNF, TNF, NTF3, NEFL, NGF
GO:0042742	Defense response to bacterium	4	2.34 × 10^−4^	TNF, STAB1, IFNB1, IL10
GO:0048666	neuron development	5	4.25 × 10^−4^	BDNF, NTF3, NTNG2, NEFL, NGF
GO:0043066	Negative regulation of apoptosis	5	5.01 × 10^−4^	BDNF, TNF, NEFL, IL10, NGF
GO:0043069	Negative regulation of programmed cell death	5	5.28 × 10^−4^	BDNF, TNF, NEFL, IL10, NGF
GO:0060548	Negative regulation of cell death	5	5.33 × 10^−4^	BDNF, TNF, NEFL, IL10, NGF
GO:0030030	Cell projection organization	5	5.80 × 10^−4^	BDNF, NTF3, NTNG2, NEFL, NGF

**Table 2 diagnostics-12-01914-t002:** GO analysis molecular function (MF) of genes associated with cognitive dysfunction in MS.

GO ID	Description	Count	*p* Value	Genes
GO:0005125	Cytokine activity	6	1.30 × 10^−6^	IL17A, IFNA1, TNF, IFNB1, IL17F, IL10
GO:0008083	Growth factor activity	5	2.07 × 10^−5^	BDNF, NTF4, NTF3, IL10, NGF
GO:0005165	Neurotrophin receptor binding	2	0.002156	NTF3, NGF
GO:0005132	Interferon-alpha/beta receptor binding	2	0.009666	IFNA1, IFNB1
GO:0032813	Tumor necrosis factor receptor superfamily binding	2	0.032931	TNF, NGF
GO:0005200	Structural constituent of cytoskeleton	2	0.076944	NEFL, SPTB

**Table 3 diagnostics-12-01914-t003:** GO analysis cell component (CC) of genes associated with cognitive dysfunction in MS.

GO ID	Description	Count	*p* Value	Genes
GO:0005576	Extracellular region	12	1.04 × 10^−6^	IL17A, IFNA1, BDNF, TNF, NTF4, NTF3, IFNB1, IL17F, CHI3L1, NTNG2, IL10, NGF
GO:0044421	Extracellular region part	9	3.93 × 10^−6^	IL17A, IFNA1, TNF, IFNB1, IL17F, CHI3L1, NTNG2, IL10, NGF
GO:0005615	Extracellular space	8	5.43 × 10^−6^	IL17A, IFNA1, TNF, IFNB1, IL17F, CHI3L1, IL10, NGF

## Data Availability

Not applicable.

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
