# Peer review of "Integrated Approaches to Identify miRNA Biomarkers Associated with Cognitive Dysfunction in Multiple Sclerosis Using Text Mining, Gene Expression, Pathways, and GWAS"

_diagnostics, 2022, doi:10.3390/diagnostics12081914_

Round 1

Author Response

This is a very interesting study, but the pathways analysis is lacking of a cornerstone element, i.e. the cerebral venous drainage and the related transcriptomics in multiple sclerosis. Cortical and Deep Gray Matter Perfusion is linked with Cognitive Performance in Multiple Sclerosis Patients. Front Neurol. 2020 Jul 17;11:700. doi: 10.3389/fneur.2020.00700. Moreover, perfusion is negatively influenced by the impairment of the cerebral venous drainage and hypoperfusion of brain parenchyma is associated with reduced jugular flow in multiple sclerosis Mult Scler Int. 2013;2013:598093. doi: 10.1155/2013/598093. Venous abnormalities are very common in MS Phlebology. 2015 Mar;30(2):98-104. doi: 10.1177/0268355513515473, and jugular flow is strongly related to cognitive disorders J Alzheimers Dis. 2021;84(2):787-796. doi: 10.3233/JAD-210490. PMID: 34602471.

On the other hand, fixing the jugular flow improves brain perfusion. J Vasc Surg Venous Lymphat Disord. 2016 Oct;4(4):434-45. doi: 10.1016/j.jvsv.2016.06.006.
Finally, transcriptomics of the jugular vein in MS shows several points of contacts with your findings Mol Med. 2018 Aug 9;24(1):42. doi: 10.1186/s10020-018-0043-4.

Int J Mol Sci. 2021 Dec 28;23(1):310. doi: 10.3390/ijms23010310.

For the reasons above, the role of perfusion and of venous abnormalities in MS and in cognitive function needs to be amended, as well as the Authors have to compare the expression profiles of the jugular veins reported in the last two papers with their own findings.

Our reply

Thanks for your suggestions related to our pathway analysis. As suggested “the pathways analysis is lacking a cornerstone element, i.e. the cerebral venous drainage and the related transcriptomics in multiple sclerosis” is useful, we studied the use of cornerstone element in our study.

As we mainly focus on the miRNAs associated with MS cognition, we couldn’t find their relation to the pathways at this point of time. Hence, we couldn’t compare our findings with the following research works.

“Mol Med. 2018 Aug 9;24(1):42. doi: 10.1186/s10020-018-0043-4.

Int J Mol Sci. 2021 Dec 28;23(1):310. doi: 10.3390/ijms23010310.”

 However, we have incorporated the importance and the information regarding the jugular veins in our introduction section.

Reviewer 2 Report

This is an interesting study using biomarkers related to  MS cognitive impairment and potential treatments being used. Using miRNAs to the biological pathways for risk in MS can help health care providers in genetic counseling of those who can provide such data. But in addition, one might be able to target therapies according to the profile of the particular person with MS.

The analysis of correlating key functional genes appears to be managed well and well presented. I feel the readership will enjoy reading this manuscript and can build on the analysis presented.

Author Response

This is an interesting study using biomarkers related to  MS cognitive impairment and potential treatments being used. Using miRNAs to the biological pathways for risk in MS can help health care providers in genetic counseling of those who can provide such data. But in addition, one might be able to target therapies according to the profile of the particular person with MS.

The analysis of correlating key functional genes appears to be managed well and well presented. I feel the readership will enjoy reading this manuscript and can build on the analysis presented.

Our reply

Thanks for the positive comments of our manuscript.

Reviewer 3 Report

Herein, the authors used text mining, GWAS signals and pathway analysis to identify potent biomarkers associated with multiple sclerosis. The results of the presented study sound interesting. Some further issues could make the manuscript more consistent and well-founded as the authors should clarify some point to further strengthen their conclusions:

1.       Can the authors construct a title telling what is the major conclusion? The word “investigation” tells a procedure. Consider rewriting the sentence for clarity.

2.       Technically, the abstract could be slightly revised; it seems like highlights, not a structured section. Introduction should also be strengthening giving also some insights related to the following study.

3.       Wording should be checked carefully (for examples lines 116, 127, 346)

4.       In general, figures quality improvement and better results presentation is needed (at least Figures 1 and 3).

5.       In Table 1 please add abbrevations (i.e: FDR, false discovery rate). Also stress the significance of FDR values.

Author Response

Herein, the authors used text mining, GWAS signals and pathway analysis to identify potent biomarkers associated with multiple sclerosis. The results of the presented study sound interesting. Some further issues could make the manuscript more consistent and well-founded as the authors should clarify some point to further strengthen their conclusions:

  1. Can the authors construct a title telling what is the major conclusion? The word “investigation” tells a procedure. Consider rewriting the sentence for clarity.

Thanks for your suggestion. The title of the manuscript is revised accordingly.

  1. Technically, the abstract could be slightly revised; it seems like highlights, not a structured section. Introduction should also be strengthening giving also some insights related to the following study.

As per the suggestion of the reviewer the abstract is modified. Introduction section is modified with more insights of our study and included in the manuscript.

  1. Wording should be checked carefully (for examples lines 116, 127, 346)

Words are modified in lines 116, 127, 346 and the complete manuscript is carefully rechecked and the changes are made.

  1. In general, figures quality improvement and better results presentation is needed (at least Figures 1 and 3).

Figures are improved for the quality and the results are presented with improvement. As Figure 1 is large, we moved it to supplementary data for better visualization.

  1. In Table 1 please add abbrevations (i.e: FDR,false discovery rate). Also stress the significance of FDR values.

FDR abbreviation is included in table1 and the significance of the FDR values of our study is included in the manuscript.

Round 2

Reviewer 1 Report

In the introduction section the Authors again do not explain why flow blockages of the IJVs may impact the pathophysiology. What they report is not enough for the readers. Please explain the role of drainage and perfusion on cognition in MS, as clearly requested in my first revision.

Author Response

Reply to the reviewer’s comments:

Reviewer #1:

In the introduction section the Authors again do not explain why flow blockages of the IJVs may impact the pathophysiology. What they report is not enough for the readers. Please explain the role of drainage and perfusion on cognition in MS, as clearly requested in my first revision.

We have modified the introduction section with the details of drainage and perfusion on cognition in MS and incorporated in the manuscript (pp. 2, paragraph 3).
